# Dietary Fiber and Gut Microbiota in Renal Diets

**DOI:** 10.3390/nu11092149

**Published:** 2019-09-09

**Authors:** Carla Camerotto, Adamasco Cupisti, Claudia D’Alessandro, Fulvio Muzio, Maurizio Gallieni

**Affiliations:** 1Luigi Sacco Hospital, ASST Fatebenefratelli Sacco, 20157 Milano, Italy; 2Department of Clinical and Experimental Medicine; University of Pisa, Pisa 56126, Italy; 3Department of Biomedical and Clinical Sciences “Luigi Sacco”, Università di Milano, 20157 Milano, Italy

**Keywords:** renal diets, fiber, renal nutrition, chronic kidney disease, gut microbiota

## Abstract

Nutrition is crucial for the management of patients affected by chronic kidney disease (CKD) to slow down disease progression and to correct symptoms. The mainstay of the nutritional approach to renal patients is protein restriction coupled with adequate energy supply to prevent malnutrition. However, other aspects of renal diets, including fiber content, can be beneficial. This paper summarizes the latest literature on the role of different types of dietary fiber in CKD, with special attention to gut microbiota and the potential protective role of renal diets. Fibers have been identified based on aqueous solubility, but other features, such as viscosity, fermentability, and bulking effect in the colon should be considered. A proper amount of fiber should be recommended not only in the general population but also in CKD patients, to achieve an adequate composition and metabolism of gut microbiota and to reduce the risks connected with obesity, diabetes, and dyslipidemia.

## 1. Introduction

Chronic kidney disease (CKD) is a growing public health problem, affecting about 10% of the population worldwide, with diabetes, hypertension, and obesity the most important risk factors for its occurrence in developed countries [1]. The importance of nutrition in a nephrology setting has been recognized as crucial for the management of CKD to slow down disease progression and to correct symptoms. The mainstay of dietary treatment of renal patients has been protein restriction coupled with adequate energy supply to prevent malnutrition and with the correct management of electrolytes abnormalities [2]. Besides, renal diet composition may also influence gut microbiota, which has been proved to play a role in reducing toxins production and preserve renal function, slowing CKD progression [3,4,5,6]. Several aspects of renal diets, including fiber content, can modulate the gut microbiota metabolism of CKD patients [7]. Nutritional counseling can help individuals limit dietary phosphorus and potassium load while maintaining or increasing dietary fiber intake [8,9].

This paper aimed to summarize the latest literature on the role of different types of dietary fiber in CKD, with special attention to gut microbiota and the potential protective role of renal diets.

## 2. Fiber Definition and Recommendation

The research about the use of dietary fiber and human health is large and heterogeneous. Since the 1970s in Western countries, recommendations of scientific societies [10,11,12,13] for the general population have suggested an adequate intake of fiber for a healthy diet (Table 1). The nutrition nephrology guidelines KDOQI (Kidney Disease Outcomes Quality Initiative) [14], KDIGO (Kidney Disease Improving Global Outcomes) [15], and EBPG (European Best Practice Guidelines) [16] do not provide specific guidance for CKD patients. Beto et al. [17] indicated, for all CKD stages, the same amount suggested for the general population.

In the general population, the current guidelines recommend a total fiber (both soluble and insoluble) intake of 20–35 g/day [10]. The mean intake of dietary fiber in the United States is 17 g/day with only 5% of the population meeting the adequate intake [18].

Regulatory Authorities recommend a proper fiber intake to assess benefits on stool excretion for laxation and a regular intestinal transit, as well as for its metabolic action in maintaining proper cholesterol and glucose metabolism [19]. The role of fiber in health and disease is more complex, and the modern analytic techniques open the way to the complex interaction on human metabolism connected with the fiber intake. Beyond the total quantity of fiber to be consumed, no indications are specifying the types of fiber or the proportions of the different fiber-containing food to reach an optimal intake.

An appropriate definition of dietary fiber is essential [20]. In 2007, FAO/WHO experts underscored that the term dietary fiber is referred to as nondigestible carbohydrates contained in grains, seeds, vegetable, and fruits [21]. This is the most internationally accepted definition. The US Codex Alimentarius Commission [22], which sets international guidance standards for food, introduced the following definition of dietary fiber in 2009: carbohydrate (CHO) polymers with ten or more monomeric units, which are not hydrolyzed by the endogenous enzymes in the small intestine of humans and belong to the following categories: (a) edible CHO polymers, naturally occurring in the food as consumed; (b) CHO polymers, obtained from food raw material by physical, enzymatic, or chemical means; and (c) synthetic CHO polymers [22]. It was also indicated that individual countries could decide whether they accept oligosaccharides with a degree of polymerization (DP) from three to nine (included) as being fiber, raising some uncertainty regarding a correct and univocal definition.

Dietary fiber is made up of carbohydrates polymers with three or more monomeric units (MU), plus associated substances. Table 2 presents the classification of dietary fiber according to chemical structure. Other physicochemical characteristics like water solubility, viscosity, and fermentability have been considered [23].

In the European Union (EU), the minimum number of carbohydrates in fibers is three (MU ≥ 3).

To be included in the fiber definition, a health benefit is required, such as “decrease of intestinal transit time, an increase of stool bulk”, “reduction in blood cholesterol”, and “modulation of glucose metabolism” [24]. The adequate intake for the US was based on the cholesterol-lowering effect. The European Food Safety Authority (EFSA) indications on health claims related to dietary fiber report the use of the term “soluble” and “insoluble” in the literature to classify dietary fiber according to a physicochemical characteristic linked to different physiological effects [25]. However, aqueous solubility does not always predict physiological effects, so a different classification according to the main characteristic of viscosity, fermentability, and bulking effect in the colon, which is related to water solubility and swelling capacity, has been proposed [23]. There is an overlap between the characteristics used for classification, creating confusion in correlating specific dietary fiber characteristics to observe health outcomes (Table 3).

Solubility refers to dissolution in water, but it is the viscosity (capacity to gel with water) of certain soluble fibers that influences chyme consistency and slows enzymatic digestion of consumed nutrients to absorbable components. Oligosaccharides are highly soluble, and fermentable fibers include fructooligosaccharides (FOS) and galactooligosaccharides (GOS). These short-chain fibers are highly fermentable due to their small size and solubility.

Soluble, non-viscous, readily fermentable fibers (inulin, wheat dextrin) dissolve in water and are rapidly and completely fermented [26]. Soluble, viscous, readily fermentable fibers (β-glucan, gums, pectin) are similar but form a gel-like consistency with water [27]. These characteristics are then lost following fermentation. Soluble, viscous, slowly fermented fibers (psyllium) also form a gel-like consistency but do not undergo extensive fermentation. As such, the capacity to interact with water is preserved throughout the colon. This allows softening of stools in those suffering from constipation and adds firmness to loose stools. Insoluble fibers (wheat bran, lignin, cellulose) exert a laxative effect by stimulation and irritation of gut mucosa to increase secretion and peristalsis [28].

EFSA included in the definition of dietary fiber also polymers obtained by physical, enzymatic, or chemical means with a demonstrated beneficial physiological effect [25]. The EU Regulation (EC) No. 1924/2006 on nutrition and health claims for foods specifies the requirement for the use of the terms “source of fiber” or “high in fiber”. The claim that a food is a source of fiber may only be made if the product contains at least 3 g of fiber per 100 g or at least 1.5 g of fiber per 100 kcal (418 kJ). The claim that a food is high in fiber, and any claim likely to have the same meaning for the consumer, may only be made if the product contains at least 6 g of fiber per 100 g or at least 3 g of fiber per 100 kcal (418 kJ) [29].

## 3. Dietary Fiber and Chronic Kidney Disease

The study of fiber and other dietary components is complex, not only for different proprieties of different kind of fibers. First, it is difficult to assess the dietary fiber intake because no biomarkers are available, and dietary histories are often unreliable. Then, the interaction of fiber with the individual human gastrointestinal tract can change due to previous dietary fiber habits, gut microbiota differences in various individuals, and previous disease or medication [30]. Fiber intake has been investigated as a dietary aspect that may exert a favorable action on renal health through different mechanisms (Figure 1).

### 3.1. Role in Intestinal Transit

The bulking effect is important to maintain a proper intestinal transit. In Europe, constipation is one of the most common gastrointestinal complaints; the causes are variable, including medication effects, increasing age, lifestyle, and dietary habits [31]. The bulking effect is dependent on the water-binding capacity of dietary fiber and on fermentation, which alters the osmotic balance and increases fecal biomass. Fiber from wheat bran has been shown to have a high bulking effect (5.4 g stool weight increase per 1 g of wheat fiber) due to its resistance to fermentation [18]. In a cohort study conducted in 3 million US Veterans, a severe constipation status was correlated with an increased risk of CKD and a faster decline of glomerular filtration rate [32]. In CKD, medications like phosphate binders or antibiotics, water reduction, and dietary restriction due to potassium management may exacerbate constipation [33,34]. Constipation rates reach 63% in hemodialysis patients and 29% in peritoneal dialysis [35]. Considering that intestinal transit time is increased in advanced kidney failure, an increase in dietary fiber should benefit CKD patients. In addition to reducing constipation, this effect could improve the intestinal microbial profile and increase concentrations of fermentation products [36].

### 3.2. Role in Weight Control

Similar viscous types of fiber (such as guar gum, pectins) are associated with reduced appetite and better control of body weight [37]. A diet rich in fiber is characterized by the presence of whole cereals, legumes, fruits, vegetables, and a low-density caloric amount.

Observational evidence for the effects of different sources or types of dietary fiber on body weight management is rather limited and inconsistent in terms of strength of association. There is some evidence from pooled data from five EPIC (European Prospective Investigation into Cancer and Nutrition) centers study that individuals with higher total and grain fiber intakes experienced smaller annual weight gains. Over the 6.5-years follow-up, for each 10 g greater intake of total fiber, the mean weight gain was lower by 39 g/year in the 89,000 European participants. This small annual improvement might potentially contribute to significantly greater lifetime weight stability in higher fiber consumers [38]. Obesity is an important risk factor for CKD, and healthy body weight throughout the life, as a result of education to a proper diet and lifestyle, is a public health goal and should be recommended to prevent kidney disease [39].

A reduced appetite can be a harmful effect in patients with end-stage kidney disease when symptoms of uremia as loss of appetite or nausea can negatively affect their nutritional status. Salmean et al. [40] conducted a pilot study to test the impact of a diet with foods added in fiber on appetite in adults with CKD. Participants were provided with control foods (cookies, snack bars, and breakfast cereal) containing <2 g/day of fiber for 2 weeks, followed by similar foods providing 23 g/day (pea hull, inulin, and resistant corn dextrin) for 4 weeks, to incorporate into their usual diets. No change in body weight or energy intake occurred over the six-week study. Scores for appetite did not change, but suggested poor appetite in seven participants, with or without added fiber, and a significant risk of at least 5% weight loss within six months. Low food intake is an important risk factor for malnutrition and adverse outcomes in patients with CKD. Poor appetite is common with progression of renal damage, and the impact of a diet rich in fiber on appetite, with or without supplements, justifies further investigation at a different stage of CKD [40].

### 3.3. Role in Cancer Prevention

Kidney cancer is among the 10th most common presenting cancers in the Western world, with lifestyle and dietary habits likely playing a significant etiological role [41]. A recent review by Huang et al. [42] summarized the evidence from two cohorts and five case-control studies in a meta-analysis on dietary fiber intake and risk of renal cell carcinoma (RCC). When comparing the highest against the lowest dietary fiber consumers, the pooled estimate of risk for renal cell carcinoma indicated that the total dietary fiber intake was associated with reduced RCC risk (RR 0.84, 95% CI 0.74–0.96). The study also revealed some differential associations according to the source of dietary fiber (greatest risk reductions for fiber from legume and vegetable sources, rather than fiber from grains or fruit). However, using a dose-response meta-analysis approach, the authors were unable to report any evidence of diminishing risk with increasing intakes of dietary fiber. This points to the need for further, large prospective cohort studies to explore potential links between dietary habits and kidney cancer [42]. The World Cancer Research Fund analysis of the evidence on cancer prevention and survival recommends a diet rich in whole grain cereals, fruits, and vegetables [43].

### 3.4. Role in Glucose and Lipid Metabolism

Diabetes is a growing worldwide epidemic, and CKD is a debilitating complication. Cardiovascular disease (CVD) is a frequent cause of mortality in patients with DM, and CKD is a risk factor for CVD [14,15].

There is enough evidence to support several EFSA health claims that certain types of fiber, including β-glucans and pectins (viscous, soluble, and fermentable), if consumed within a meal, may contribute to the reduction of blood glucose rise after a meal [26]. Psyllium (viscous, soluble but not fermentable), which delays degradation and absorption of nutrients, can reduce total glucose and cholesterol absorption [23,44].

The cholesterol-lowering effect depends on increased viscosity of fiber, reducing the re-absorption of bile acids, increasing the synthesis of bile acids from cholesterol, and reducing circulating LDL-cholesterol concentrations [28].

Evidence supports a reduced risk for type 2 diabetes with additional fiber daily consumption. Both soluble and insoluble fibers are associated with improved outcomes. The American Diabetes Association (ADA) recommends that patients with DM should consume at least 14 g of fiber for each 1000 Kcals daily, and it suggests that carbohydrate intake from vegetables, fruits, legumes, and whole grains, with an emphasis on foods higher in fiber and lower in glycemic load, is preferred over other sources of carbohydrates [10]. The American Heart Association (AHA) also endorses healthy dietary patterns rich in fiber to prevent CVD, such as Dietary Approaches to Stop Hypertension (DASH) and Mediterranean diets [11].

A recent review analyzed the role of dietary fiber in diabetic kidney disease [45]. Starting from 1814 studies, 48 articles were evaluated, and due to lack of dietary or renal outcome information, only seven clinical trials were included. Moreover, only two of them provide information about the kinds of fiber used, soluble or insoluble. Renal outcomes were albuminuria, proteinuria, changes in estimated glomerular filtration rate, dialysis. A vegetarian diet showed a beneficial effect on renal outcomes, but the individual effect of fiber intake could not be evaluated. Limitation for the small numbers of studies, for the short term follow-up, and for the amount of fiber, which was lower than recommended by most guidelines, led to the conclusion that further investigation is needed to deliver evidence that is currently still limited [45].

### 3.5. Role in the Gut Microbiome

Dietary fiber is the key substrate for microbial growth of bacteria living in the gut (prebiotic action). The gut microbiota is a true ecosystem made up of an estimated number of 3.8 x 10^13^ microorganisms living throughout the human body of a 70 kg individual [46]. Thus, the number of bacteria in the body is of the same order as the number of human cells (3.0 × 10^13^), and their total mass is about 0.2 kg. The colonization is established in the first years of life, reaching the maximum complexity in adulthood, when the dominant species are Bacteroides, Firmicutes, and Actinobacteria [47].

Fermentable fibers, such as oligosaccharides, β-glucans, gums, some hemicelluloses, and some resistant starches, are the substrate for bacteria metabolism producing short-chain fatty acids (SCFAs), primarily acetate, propionate, and butyrate [26]. Recent evidence has supported the role of these metabolites in the regulation of immunity, blood pressure, glucose, and lipid metabolism. These effects may represent the link between microbiota and host homeostasis [48]. More than 90% of SCFAs are absorbed by colonic epithelial cells, providing an important energy source [49]. Due to fermentation, fiber provides an energy intake of 2 kcal per gram [15]. Butyrate is rapidly used as an energy source by colonocyte; most of acetate and propionate enters the portal circulation and peripheral blood [50]. SCFAs reach the bloodstream and act on the immune system by modulating inflammatory gene expression, chemotaxis, differentiation, proliferation, and apoptosis [51]. These beneficial effects are related to their property as histone acetylation inhibitors and activation of transmembrane cognate G protein-coupled receptors [52]. An important role of fiber is the trophic action on intestinal epithelial cells, a physical barrier against the entrance of pathogenic microorganisms or their molecules into the portal circulation. Changes in the intestinal barrier with an increased permeability are common in CKD. SCFAs may contribute to kidney health by reducing stimuli to systemic inflammation, maintaining an intact mucosal barrier, both modulating the immune system and the anti-inflammatory response [51].

Data from 14,543 participants in the National Health and Nutrition Examination Survey III (NHANES III) showed for each 10 g/day increase in total fiber intake (total, soluble, and insoluble), the odds of elevated serum C-reactive protein levels were decreased by 11% and 38% in those without and with kidney disease, respectively [53]. The authors concluded that taken together these data suggest a stronger role of dietary fiber in lowering inflammation in the CKD population and this is a likely potential mechanism for the association of higher dietary fiber with lower mortality. Unfortunately, according to NHANES III data, the average dietary fiber intake in the CKD population was about 15.4 g/day, which was much lower than recommended [53].

### 3.6. Role in Biomarkers of Renal Function

A dysbiosis of the microbiota appears to be a risk susceptibility factor for the development of kidney disease, following injury or in predisposed individuals. The progressive reduction in kidney function significantly contributes to worsening the intestinal dysbiosis [54,55]. Therefore, the imbalance in gut microbiota contributes to the accumulation of gut-derived uremic toxins [56]. Toxic gases, indoxyl sulfate, p-Cresyl sulfate, amines, ammonia, and trimethylamine n-oxide (TMAO) as well as products of bacterial lysis translocation, i.e., precursors for lipopolysaccharides (LPS), may be absorbed into the bloodstream and be responsible for systemic inflammation. Indeed, several studies have shown that these toxins are reliable markers of cardiovascular disease and mortality in CKD patient [47,57]. The increase in bowel transit velocity, the reduction of constipation, and changes of the intestinal bacterial composition may reduce serum urea by altering the urea enterohepatic cycling, increasing nitrogen excretion via fecal mass [58]. Because urea is diffusible through the colon and creatinine is metabolized by intestinal bacteria, their net endogenous production and hence their retention can be potentially reduced in CKD patients. This phenomenon could also represent a bias for the estimation of residual renal function. Data about dietary fiber effects in CKD using serum creatinine and urea have been reviewed by Chiavaroli et al. [59]. This systematic review and metanalysis identified 8051 reports, but it included only 14 controlled feeding trials that met the eligibility criteria for analyses, involving 143 participants (median number of participants: 9, range 3–22) with a median age of 51.9 years. Though no data were reported about the CKD stage and fiber intake at baseline, the study is relevant because the supplemented median fiber dose, about 27 g/day, is consistent with the recommendation for adults. Results indicate a reduction in serum urea and creatinine levels associated with dietary fiber intake, which occurred in a dose-dependent manner for creatinine. However, it should be kept in mind that accumulation of urea and creatinine do not directly induce renal damage. In most studies, fiber was supplemented as fermentable fiber type (psyllium, gum Arabic, inulin, and lactulose), yielding a total median fiber dose of 26.9 g/day, with a very high range of 3.1–50 g/day. The incidence of CKD assessing e-GFR was evaluated by the Tehran Lipid and Glucose Study [60] that followed 1630 participants, mean age 42.8 years, for 6 years, who were initially free from CKD. Data about dietary fiber was collected with a valid and reliable food frequency questionnaire. The authors observed a reduced risk of incident CKD in higher tertiles of fiber intake (average 36.6 g/day). The study reported the distribution of fiber from different foods (fruits, vegetables, cereals, and legumes). With every 5 g/day increase in total fiber, the risk of incident CKD decreased by 11%; a protective association was observed for vegetables and legumes fiber. Odds ratios for participants in the highest compared with the lowest tertile of fiber from vegetables was 0.63 (95% CI 0.43–0.93), and, from legumes, it was 0.68 (95% CI 0.47–0.98) [60].

## 4. Renal Diets

The diets for CKD patients are characterized by the control or reduction of protein intake, the increase of carbohydrates supply, and the increase of plant-origin protein and food with respect to animal-origin foods. In other words, with respect to the current western diet pattern, diets implemented in renal patients favor complex carbohydrate and fibers with respect to animal-origin proteins and refined foods [61].

All these aspects may have several favorable effects on gut microbiota metabolism. One of the effects of a successful dietary protein restriction is the reduced nitrogen waste products retention, the most common biomarker being urea. In CKD, the accumulation of urea in body fluids causes its diffusion into the intestinal lumen, where it is converted into ammonia by the urease-positive species, and finally hydrolyzed to ammonium hydroxide [7]. The latter causes changes of the tight junctions, damage to the epithelial barrier, and an increase in intestinal permeability, resulting in the passage into the bloodstream circulation of bacterial toxins [62]. Consequently, the activation of local and systemic chronic inflammatory mechanisms induces further damage to the intestinal epithelial barrier, triggering a vicious circle that also favors the progression of renal damage [63]. Through saccharolytic fermentation in the intestinal lumen, carbohydrates are converted to SCFAs, such as acetate, butyrate, and propionate, which have anti-inflammatory and protective effects on immune function and intestinal barrier integrity. On the other hand, the products of proteolytic fermentation, such as phenols, indole, amines, and ammonium, are potentially toxic metabolites and reduce the circulating levels of SCFAs [56]. Complex carbohydrates, such as dietary fiber and fructo-oligosaccharides (FOS), obtained by hydrolysis of plant-origin inulin, are today recognized as prebiotics, that is substances able of favorably modifying the composition of gut microbiota, stimulating growth, and metabolic activity of beneficial microorganisms, such as Bifidobacteria and Lactobacilli with saccharolytic metabolism [57,64,65]. Recent data showed that serum concentrations of p-cresol and indoxyl sulfate were reduced by oral p-inulin intake in CKD patients on hemodialysis [7].

Therefore, the diet may have a potential impact on modifying the composition of the microbiota and optimizing homeostasis in patients with CKD. Low-protein renal diets remain crucial in the therapy of patients with CKD [66]. Diets with a very low protein and vegan content reduce the intake of substrates for protein fermentation and ensure a high intake of dietary fiber, which increases the transit speed in the colon, decreasing the production and absorption of uremic toxins [2]. Patients with CKD, not yet on dialysis, switched from a mild dietary protein restriction to a strict protein restriction (0.3 g/kg body weight/day) with supplementation of essential amino acids, and keto-analogs (KA) showed a reduction of 37% in the blood concentration of indoxyl sulfate, achieving at the same time an adequate protein metabolism and a low production of urea [67]. In a retrospective, propensity score-matched study of CKD patients who received more than 3 months of a KA supplemented low-protein diet during the year preceding the start of dialysis, a lower long-term risk of all-cause mortality (hazard ratio (HR) 0.77, 95% CI 0.70–0.84), major cardiac and cerebrovascular events (HR 0.86, 95% CI 0.78–0.94), and infection-related death (HR 0.76, 95% CI 0.67–0.87) were observed compared to patients on a different non-specified diet [68]. Bellizzi et al. [69] confirmed that a very low-protein diet supplemented with amino acids and KA during CKD does not increase mortality in the subsequent renal replacement treatment period. Similarly, in selected (14% of screened patients) compliant patients following a vegetarian keto-analog–supplemented very low–protein diet, which is rich in fiber, it was shown that this kind of diet is nutritionally safe and could defer dialysis initiation [70].

Dietary interventions, consisting of the increase of complex carbohydrates and fibers intake with a reduction of animal proteins and refined and processed foods, are potentially useful in the initial phases of CKD to reduce uremic toxins production, which is associated with the progression of CKD [7,71]. This is the case of the Mediterranean diet or the DASH (Dietary Approaches to Stop Hypertension) diet, characterized by a high intake of food of vegetable origin rich in natural and fiber prebiotics and reduced salt intake, refined sugars, animal fats, and red meats [72].

Furthermore, it is useful to limit the consumption of processed food products because modern conservation processes, which have the purpose of eliminating pathogenic bacteria and guarantee the durability of food, reduce the intake of even commensal microorganisms beneficial for the intestinal flora, and because these products can constitute “hidden” sources of phosphates and sodium. Instead, in the most advanced stages of CKD, low protein diets rich in carbohydrates, and sometimes, vegetarian diets are useful options. Finally, it is known that physical activity is associated with benefits on blood pressure control, glucose and lipid metabolism, and endothelial function. A sedentary lifestyle slows intestinal transit, and preliminary evidence suggests that exercise is associated with greater changes in gut microbiota and a reduction in pathogenic components. In the patient with CKD, exercise is an anabolic stimulus that integrates nutritional interventions to counteract the loss of lean mass, positively influencing nutritional status and quality of life. Therefore, in the management of CKD patients at any stage, regular physical activity should be promoted as an integral part of the nutritional plan [73].

## 5. The Fiber in Renal Diets

The analysis of nutrients content of four different types of diets for renal patients showed an average fiber amount of 7.66 g/1000 Kcal for conventional low protein diet (0.6 g protein/kg), 16 g/1000 Kcal of low-protein vegan diet (0.6 g protein/kg), 11.6 g/1000 Kcal for very low protein diet (VLPD), and 10.4 g/1000 Kcal for a 0.8 g protein/kg diet [74].

In Italy, an animal-based low protein diet includes the use of special low protein foods, formulated to give a high number of calories with negligible content of nitrogen, potassium, and phosphorous. In other words, the proteins in the diet are mostly from the animal source because cereals and legumes are excluded and substituted by high-energy protein-free products. These are starch-made substitutes of regular bread, pasta, and bakery products, supported by a national health system, with a formulation high in technological value that allows a similar taste to traditional products [66,75]. Fibers are already present in special protein-free foods, baked products, in particular, to mimic the structural role of gluten. The most used fibers are carob, xanthan gum, and guar cellulose and its derivatives [76,77]. Protein-free pasta has a low content of fiber because a specific rout of production is used for its production that does not need the addition of specific ingredients. Currently, protein-free products have been further enriched with fibers. Fibers content have been increased significantly in all products, including pasta and baked products, not only for a technological reason but also for the well-known metabolic role of fibers, in particular regarding gut microbiota composition and metabolism.

Comparing the nutritional facts labels of protein-free products available about 10 years ago and the current ones, a relevant increase in fibers content emerges. The average content of fiber in bread increased from 4.2 to 10.8 g per 100 g, from 1.5 to 4.8 g per 100 g in pasta, and from 0.8 to 3.3 g per 100 g in biscuits and cakes. Instead, the fiber content in protein-free flour is quite the same.

Table 4 reports the fiber content of the main categories of protein-free products; the data resulted from the five better-known brands of protein-free products. Not surprisingly, the fiber content in protein-free products is higher than that of regular products. For example, the average content of fiber for regular pasta is 2.7 g/100 g versus 4.8 g/100 g of protein-free pasta. The average content of fiber in regular bread is 2.7 g/100 g versus 10.8 g/100 g of protein-free bread that is even higher than that of whole-grain bread, 6.5 g/100 g (fiber content of regular products comes from the National Institute for Food and Nutrition database) [78]. It is quite interesting to observe also that the types of fiber currently used seem to be more natural with respect to the past (Table 4).

Achieving higher fiber intake remains a concern in the renal diet because of increased potassium and phosphorus levels. Foods added in fiber or supplements may be a useful option for the high energetic amount and the reduced content in electrolytes. Remarkably, fiber supplementation in CKD patients was shown to lower plasma p-cresol by 20% to 37% (in participants with elevated compliance) [6].

A low protein diet with an increased dietary fiber content might add additional benefit in the rate of progression of CKD, but the causal relationship in preserving kidney function could be difficult to establish between higher dietary fiber and the low protein diet. Whether dietary fiber per se or the other nutrients that are present in the foods that are high in fiber decrease systemic inflammation is still an open question. Fruits and vegetables contain vitamins and antioxidants, making these foods necessary for a healthy diet, and natural sources are to be preferred [79]. Bioavailability of phosphorus in fruits, vegetables, and whole grains is lower when compared to processed food [80]. To reach enough dietary fiber and to ensure a low potassium content in the diet for patients with advanced CKD, vegetables should be cooked by boiling in water before ingestion and fruit should be properly selected [61,81].

Restoring fiber intake after a prolonged period of dietary deficiency represents a significant challenge, not simply educational but also physiological. “Addition of” or “changes in” fiber intake leads to bloating, abdominal cramps, and increased flatulence. Furthermore, delayed gastric emptying and digestion, from soluble and viscous fibers, may aggravate symptoms of dyspepsia. These unwanted symptoms are associated with many gastrointestinal and functional disorders and may affect adherence to adopt a high fiber diet [28].

Considering the association of fiber and hard outcomes, high dietary total fiber intake has been linked with better kidney function and a lower risk of inflammation and mortality in CKD patients of North America [53] and Northern Europe [82].

## 6. Fiber in Dialysis

The effects of fiber intake in dialysis patients have been assessed and reviewed by Sirich and collaborators [83,84]. In dialysis patients, an increase in dietary protein intake is suggested by current guidelines, between 1.1 g/kg/day [85] and 1.2 g/kg/day [16]. This may lead to increased production of protein-derived uremic toxins produced by the colon microbiota, which may be responsible for symptoms and increased cardiovascular morbidity. This effect may be offset by an increased fiber intake, as demonstrated by a study in hemodialysis patients, who showed significantly reduced plasma levels of indoxyl sulfate after an increase in dietary fiber in the form of high amylose corn starch for 6 weeks [83]. Another group [86] previously showed that increasing dietary fiber (oligofructose-enriched inulin for 4 weeks) reduced the production of p-cresol in hemodialysis patients. The proposed protective mechanism of fiber is related to increased delivery of SCFAs to the colon, providing energy and allowing microbes to incorporate dietary protein for growth rather than producing uremic toxins from protein breakdown [87].

Among nondiabetic peritoneal dialysis patients, fiber intake was inversely associated with all-cause mortality, with a 13% reduction of mortality for each 1 g/day of increased fiber intake [88]. Moreover, in another recent prospective cohort study of 219 prevalent hemodialysis patients, dietary fiber intake was associated with less inflammation, less myocardial hypertrophy, injury, and a reduced risk of major adverse cardiovascular events [89].

Currently, there is no indication on the ideal fiber intake for hemodialysis patient, but the estimated intake is only 11–12 g/day compared to the suggested intake for the general population of 25 g/day for women and 38 g/day for men [84].

## 7. Conclusions

Although no clear-cut evidence so far can prove the direct effect of dietary fiber in retarding the progression of CKD, and the possible adverse effects of a fiber-rich diet in end-stage CKD, such as dyspepsia, may offset its potential benefits, a proper amount of fiber could be recommended not only in the general population but also in people affected by CKD, to maintain an adequate composition and metabolism of gut microbiota and to reduce the risks connected with obesity, diabetes, and dyslipidemia. Fibers are a diversified family of compounds, classified considering viscosity, fermentability, and bulking effect in the colon, which is related to aqueous solubility. The main sources of dietary fiber are the cereal grains, starchy endosperm, fruits and vegetable cell walls, green banana, psyllium, legumes, guar gum, beetroot, rice endosperm, chicory root, Jerusalem artichoke, onion, polymers derived by hydrolysis from polysaccharides, red algae, and fungi. Dietary fiber can modulate the gut microbiota metabolism of CKD patients, reducing the bacterial generation of uremic toxins and potentially improving CKD progression and, even in dialysis patients, improving uremic symptoms and comorbidities. More clinical studies need to be conducted to assess the potential multifactorial benefits of a natural diet rich in fiber in CKD and define the impact of quality and quantity composition in different kinds of fiber.

## Figures and Tables

**Figure 1 nutrients-11-02149-f001:**
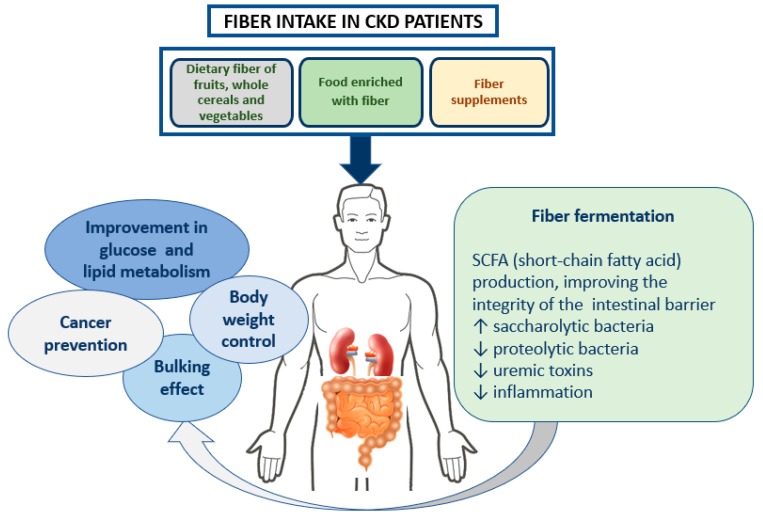
Potential beneficial effects of fiber in renal health. Fiber intake and fermentation are of crucial importance in the kidney-intestine axis, both for kidney health and for outcomes associated with other relevant conditions, such as cancer, diabetes, obesity (for references, see text). CKD: chronic kidney disease.

**Table 1 nutrients-11-02149-t001:** Guidelines recommendations on fiber intake in the general population. No specific recommendation is provided for CKD (chronic kidney disease) patients by the available nephrology guidelines.

ADA (2014): 14 g/1000 kcal or 25 g/day women, 38 g/day men [10] AHA (2014): Rich in fiber [11] USA Guideline (2015): 14 g/1000 kcal [12] EFSA European Guideline (2010): 25 g/day [13]

Legend: ADA = American Dietetic Association; AHA = American Heart Association; EFSA= European Food Safety Authority.

**Table 2 nutrients-11-02149-t002:** Classification of dietary fiber according to main chemical components, main food sources, and intrinsic characteristics (data derived from Stephen et al. [23]).

Subgroup	Class of Poly, Oligosaccharides	Main Sources	Water Solubility	Viscosity	Fermentability
No Starch Polysaccharides,MU ≥ 10	Cellulose	Outer layers of cereals.	-	-	+
	Hemicellulose	Starchy endosperm and outer layers of cereals, fruits, and vegetable cell walls.	- (soluble after alkaline extraction)	-	-
Seed (psyllium).	++	+++	+
Cereal grains (β-glucans)	+	++	++
Mannans	Seeds, green coffee beans, aloe vera.	+	-	++
Heteromannans	Grain legumes, guar gum (galactomannans), Konjac (glucomannans).	+++	+++	++
Pectins	Fruit peel, beetroot, rice endosperm, legumes.	++	++	++
Inulin and fructans	Chicory root, Jerusalem artichoke, onion, cereal grains (2–3% *w/w*).	++	-	++
Resistant oligosaccharides, MU 3-9	α-galactosides	Polymers derived by hydrolysis from polysaccharides.	+++	-	+++
β fructooligosaccharides (FOS)

	α-galactooligosaccharides (GOS)
	β-galactooligosaccharides (TOS)
Xylo-oligosaccharides (XOS)

Arabino-xylooligosaccharides (AXOS)
Polydextrose
Resistant starch, MU ≥ 10	Type 1 - physically inaccessible starch	Type1- whole grains and legumes	-		++
	Type 2 - granular starches	Type 2- Green banana			++
	Type 3 - gelatinized and retrograded starch	Type 3- Cooled starches in cooked starchy food			++
Type 4 - chemically modified	Type 4- synthetized			+
Associated substances	Lignin, waxes, chitins	Cell walls of plants, red algae, fungi.	-	waxes ++	-
MU = monomeric units					

**Table 3 nutrients-11-02149-t003:** Main fibers physicochemical characteristics (data derived from O’Grady [28]).

	Water Solubility		Viscosity		Fermentability	
	YES	NO	YES	NO	YES	NO
Beta-glucan, Gums, Pectins, Mucilage						
Lignans						
Fructo-oligosaccharides, Galacto-oligosaccharides						
Inulin						
Psyllium						
Resistant starch, Cellulose						

**Table 4 nutrients-11-02149-t004:** The fiber content of the main categories of protein-free foods made by five of the most known brands of protein-free products. The data refer to 100 g of edible product.

	Fiber Content (g/100 g)		Type of Fibers
	Median (IQ)	Min–Max	
Bread	12.5 (7.5–13)	5.0–13	Cellulose, psyllium, apple extract, deglutinated wheat fiber
Bread substitutes	7.4 (6.7–9.0)	4.0–15	
Pasta	4.8 (2.8–5.7)	3.0–7.3	Cellulose, inulin
Biscuits and cakes	3.2 (2.0–3.6)	0.5–8.5	Bamboo fiber, pectin,
Flour (for bread)	3.0 (2.8–4.0)	2.7–5.4	Cellulose, psyllium, apple extract, deglutinated wheat fiber

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
