# Peer review of "Dietary Fiber and Gut Microbiota in Renal Diets"

_nutrients, 2019, doi:10.3390/nu11092149_

Round 1

Reviewer 1 Report

In this study, the authors reviewed latest literatures about the association across different types of dietary fiber, intestinal microbiota, and chronic kidney disease and concluded that a proper amount of dietary fiber should be recommended in patient with CKD. The strength of this article is to demonstrate the potential benefit of different kinds of dietary fiber in detail. However, since low protein diet mostly correlated with higher dietary fiber, some concerns are raised to the causal relationship in the retard of CKD progression between higher dietary fiber and low protein diet. 

1.     In Section 3.1, the authors have stated that some medications use, water and potassium restriction may exacerbate the constipation in patients with CKD. We do expect some more statement about the role of dietary fiber in chronic kidney disease.

2.     In Section 3.6, the authors stated that the imbalance of gut microbiota in patients with CKD contribute to the accumulation of gut-derived uremia toxins. We do not see a direct evidence in current reference of this article. Could the authors provide more relevant reference to prove this point?

3.     Similarly, in section 3.6, the authors demonstrated that the fiber-rich diet may reduce BUN and creatinine by increasing nitrogen and creatinine excretion in fecal mass, and thus reflect a correlated better renal function. However, unlike indoxyl sulphate and p-Cresyl sulphate, accumulation of BUN and creatinine do not induce a renal damage directly, but only reflect a decrease of renal function. Furthermore, according to the authors’ statement, we may doubt that the decrease of BUN/Creatinine is because of fecal excretion rather than renal function improvement. The authors should clarify this point.

4.     In conclusion, the authors suggested a proper amount of fiber should be recommended in CKD patients. We do believe that dietary fiber has a role in glucose control and intact intestinal mucosa. However, no convincing evidence so far can prove the direct effect of dietary fiber in retarding the progression of CKD. In addition, the possible adverse effect of a fiber-rich diet, such as dyspepsia or hyperphosphatemia, may offset its potential benefits. Thus, the authors may modify their recommendation and acknowledged these concerns.

5.     Recently, low protein diet with ketoanalgue has been shown the better outcome even after dialysis than whose were not under supplement (Nutrients. 2018 Aug; 10(8): 1035.), please acknowledge that.

6.     Please discuss the vegetarian diet in CKD and explore its relationship with fiber. In 2016, Liliana Gârneata showed that patients with Vegetarian VLPD with ketoanalgue could postpone the renal function deterioration. (J Am Soc Nephrol. 2016 Jul; 27(7): 2164–2176) Therefore, the vegetarian diet and its benefit are hot on this topic.

Minor

1.     There are some spelling errors. For example, in line 17, the word should be “assess” rather than “asses”. In line 108, “can” rather than “con”. In line 220, “NHANES” rather than “NANES”.

Author Response

Reviewer 1

In this study, the authors reviewed latest literatures about the association across different types of dietary fiber, intestinal microbiota, and chronic kidney disease and concluded that a proper amount of dietary fiber should be recommended in patient with CKD. The strength of this article is to demonstrate the potential benefit of different kinds of dietary fiber in detail. However, since low protein diet mostly correlated with higher dietary fiber, some concerns are raised to the causal relationship in the retard of CKD progression between higher dietary fiber and low protein diet.

Answer: we thank the reviewer for the acute assessment of our manuscript. We agree that it is difficult to separate the effects of a reduced protein load and that of increased fiber content in the same diet. However, we do not see the two aspects as alternative effects. Our aim is pointing out the possibility of additional benefit from an increased fiber content. The following sentence was added in section 5 (lines 407-409): “A low protein diet with an increased dietary fiber content might add additional benefit in the rate of progression of CKD, but the causal relationship in preserving kidney function could be difficult to establish between higher dietary fiber and the low protein diet.

In Section 3.1, the authors have stated that some medications use, water and potassium restriction may exacerbate the constipation in patients with CKD. We do expect some more statement about the role of dietary fiber in chronic kidney disease.

Answer: We summarized the effects of fiber in CKD in figure 1. Section 3.1 is specifically dedicated to the role of fiber on intestinal transit, not on CKD. To address the reviewer’s suggestion, we added the following sentence in section 3.1 (Page 5, lines 152-156): ”Constipation rates reach 63% in hemodialysis patients and 29% in peritoneal dialysis [Yasuda, 2002]. Considering that intestinal transit time is increased in advanced kidney failure, an increase in dietary fiber should benefit CKD patients. In addition to improving constipation, this effect could improve the intestinal microbial profile and increase concentrations of fermentation products [Flint 2012]”.

In Section 3.6, the authors stated that the imbalance of gut microbiota in patients with CKD contribute to the accumulation of gut-derived uremia toxins. We do not see a direct evidence in current reference of this article. Could the authors provide more relevant reference to prove this point?

Answer: The following is a very recent and relevant reference that we added, as required (ref. 56, page 8, line 268).  Hobby GP, Karaduta O, Dusio GF, Singh M, Zybailov BL, Arthur JM. Chronic kidney disease and the gut microbiome. Am J Physiol Renal Physiol. 2019 Jun 1;316(6):F1211-F1217.

Similarly, in section 3.6, the authors demonstrated that the fiber-rich diet may reduce BUN and creatinine by increasing nitrogen and creatinine excretion in fecal mass, and thus reflect a correlated better renal function. However, unlike indoxyl sulphate and p-Cresyl sulphate, accumulation of BUN and creatinine do not induce a renal damage directly, but only reflect a decrease of renal function. Furthermore, according to the authors’ statement, we may doubt that the decrease of BUN/Creatinine is because of fecal excretion rather than renal function improvement. The authors should clarify this point.

Answer: Although BUN changes are also influenced by hydration status and protein intake, we agree with the reviewer that creatinine changes are mainly a function of renal function, provided stable general conditions.  However, the metanalysis we reported highlights that creatinine can be metabolized in the gut. Actually, it is known that the creatinine index (i.e. an estimation of endogenous creatinine production) includes serum creatinine value that is positively related to the extra-renal creatinine elimination (namely intestinal).  Hence, we cannot exclude that changes in intestine function and microbiota composition may induce little changes in serum creatinine and BUN. Nevertheless, we recognize that the message can be misleading, and we share the concept that accumulation of BUN and creatinine do not directly induce renal damage. We changed the sentence as follows (page 8, lines 272-290):

The increase in bowel transit velocity, the reduction of constipation and changes of intestinal bacterial composition may reduce serum urea by altering the urea enterohepatic cycling, increasing nitrogen excretion via fecal mass [58]. Because urea is diffusible trough the colon and creatinine is metabolized by intestinal bacteria, their net endogenous production and hence their retention can be potentially reduced in CKD patients. This phenomenon could also represent a bias for the estimation of residual renal function. Data about dietary fiber effects in CKD using serum creatinine and urea have been reviewed by Chiavaroli et al [59]. This systematic review and metanalysis identified 8051 reports, but it included only 14 controlled feeding trials that met the eligibility criteria for analyses, involving 143 participants (median number of participants: 9, range 3-22) with median age of 51.9 years. Though no data were reported about the CKD stage and fiber intake at baseline, the study is relevant because the median fiber dose supplemented, about 27 g/ day, is consistent with the recommendation for adults. Results indicate a reduction in serum urea and creatinine levels associated with dietary fiber intake, which occurred in a dose-dependent manner for creatinine. However, it should be kept in mind that accumulation of BUN and creatinine do not directly induce renal damage.

In conclusion, the authors suggested a proper amount of fiber should be recommended in CKD patients. We do believe that dietary fiber has a role in glucose control and intact intestinal mucosa. However, no convincing evidence so far can prove the direct effect of dietary fiber in retarding the progression of CKD. In addition, the possible adverse effect of a fiber-rich diet, such as dyspepsia or hyperphosphatemia, may offset its potential benefits. Thus, the authors may modify their recommendation and acknowledged these concerns.

Answer: following the reviewer’s suggestion, the first paragraph of the conclusions was modified as follows (page 12, lines 449-454): “Although no clear-cut evidence so far can prove the direct effect of dietary fiber in retarding the progression of CKD and the possible adverse effect of a fiber-rich diet in end stage CKD, such as dyspepsia or hyperphosphatemia, may offset its potential benefits, a proper amount of fiber could be recommended not only in the general population but also in people affected by CKD, to maintain an adequate composition and metabolism of intestinal microbiota and to reduce the risks connected with obesity, diabetes and dyslipidemia”.

Also, the potential detrimental effects of fiber induced dyspepsia are indicated at page 11, lines 417-425).

Recently, low protein diet with ketoanalgue has been shown the better outcome even after dialysis than whose were not under supplement (Nutrients. 2018 Aug; 10(8): 1035.), please acknowledge that.

Answer: thank you for pointing out this missing reference. We added the following paragraph in section 4 (page 9, lines 340-346): “In a retrospective, propensity score matched study of CKD patients who received more than 3 months of a KA supplemented low-protein diet during the year preceding the start of dialysis, a lower long-term risk of all-cause mortality (hazard ratio (HR) 0.77), major cardiac and cerebrovascular events (HR 0.86), and infection-related death (HR 0.76) was observed compared to patients on a different non-specified diet [Yen CL, 2018]. Bellizzi et al. [2015] confirmed that a very low-protein diet supplemented with amino acids and KA during CKD does not increase mortality in the subsequent renal replacement treatment period”.

Please discuss the vegetarian diet in CKD and explore its relationship with fiber. In 2016, Liliana Gârneata showed that patients with Vegetarian VLPD with ketoanalogue could postpone the renal function deterioration. (J Am Soc Nephrol. 2016 Jul; 27(7): 2164–2176) Therefore, the vegetarian diet and its benefit are hot on this topic.

Answer: we agree with the reviewer. The focus of the manuscript was fiber itself and therefore the paper by Gârneata et al was not considered in the first place because it did not directly mention fibers. However, it is true that vegetarian diets are rich in fiber, and this could be one of the factors causing their favorable outcome in terms of protection of residual renal function and delay of start of dialysis.

Thus, we added the following sentence in section 4 (page 9, lines 346-348): “Similarly, in selected (14% of screened patients) compliant patients following a vegetarian ketoanalogue–supplemented very low–protein diet, which is rich in fiber, it was shown that this kind of diet is nutritionally safe and could defer dialysis initiation [Garneata 2016]”.

Minor

There are some spelling errors. For example, in line 17, the word should be “assess” rather than “asses”. In line 108, “can” rather than “con”. In line 220, “NHANES” rather than “NANES”.

Answer: these typos have been corrected. Thank you for pointing them out.

Reviewer 2 Report

Although the topic is interesting in general, I do not think that the current studies available justify the preparation of a review. Indeed, the authors also write in their conclusions that “More clinical studies need to be conducted to assess the potential multifactorial benefits of a natural diet rich in fiber in CKD and define the impact of quality and quantity composition in different kinds of fiber.”

Thus, I unfortunately do not think that this review can provide some new insights in the role of fiber in CKD/renal diets.

There are some more aspects I would like to mention.

- Although I am also not a native speaker, I think that the manuscript requires substantial language editing, as a lot of grammatical and spelling errors are present.

- Also, more references should be added to the manuscript, both for the main text and the tables.

- In general, the tables show a lot of inaccuracies.

- I do not think that it is correct to discuss if fibers can prevent CKD. They may lower the risk for the development, however, the authors often discuss a disease prevention.

- Many underlying mechanisms of dietary fibers are only superficially discussed.

- The authors mention that “the fiber content in protein-free products is higher than that of regular products.” However, for me, this is not really surprising. When you exclude one nutrient (here: protein), the concentration of the other nutrients automatically increases.

Author Response

Reviewer 2

Although the topic is interesting in general, I do not think that the current studies available justify the preparation of a review. Indeed, the authors also write in their conclusions that “More clinical studies need to be conducted to assess the potential multifactorial benefits of a natural diet rich in fiber in CKD and define the impact of quality and quantity composition in different kinds of fiber.” Thus, I unfortunately do not think that this review can provide some new insights in the role of fiber in CKD/renal diets.

Answer: We understand the reviewer’s opinion, but we respectfully point out that this manuscript (defined as a “communication” rather than a review) was prepared after an invitation and after sharing the title and its concept with the Journal Editor. Thus, we hope that it can still be considered for publication.

There are some more aspects I would like to mention.

- Although I am also not a native speaker, I think that the manuscript requires substantial language editing, as a lot of grammatical and spelling errors are present.

Answer: the manuscript has been extensively reviewed, although not by an English-speaking person. We hope it now meets the standards of the Journal.

- Also, more references should be added to the manuscript, both for the main text and the tables.

Answer: more references have been added, also following the indications of the other reviewers.

- In general, the tables show a lot of inaccuracies.

Answer: there were no specific indications to which inaccuracies the reviewer is referring to. We reviewed and corrected part of the table. We hope they are now acceptable for the reviewer.

- I do not think that it is correct to discuss if fibers can prevent CKD. They may lower the risk for the development, however, the authors often discuss a disease prevention.

Answer: following this critique and the observation of the other reviewers, we tried to more clearly the potential beneficial role of dietary fiber content on the progression of CKD (section 5). However, we also add the following sentence at the beginning of the conclusions: ““Although no clear-cut evidence so far can prove the direct effect of dietary fiber in retarding the progression of CKD and the possible adverse effect of a fiber-rich diet in end stage CKD, such as dyspepsia or hyperphosphatemia, may offset its potential benefits, a proper amount of fiber could be recommended not only in the general population but also in people affected by CKD, to maintain an adequate composition and metabolism of intestinal microbiota and to reduce the risks connected with obesity, diabetes and dyslipidemia”.

- Many underlying mechanisms of dietary fibers are only superficially discussed.

Answer: there were no specific indications to which part of the manuscript the reviewer is referring to. We believe that the mechanisms of dietary fibers in renal diets are discussed throughout the manuscript and we are certainly willing to further address specific observations, if any.

- The authors mention that “the fiber content in protein-free products is higher than that of regular products.” However, for me, this is not really surprising. When you exclude one nutrient (here: protein), the concentration of the other nutrients automatically increases.

Answer: we agree and therefore changed the text as follows (page 10, line 391): “Not surprisingly, fiber content in protein-free products is higher…”

Reviewer 3 Report

Camerotto et al. have written a commentary on the relationship between dietary fiber and gut microbiota in the prevention and management of chronic kidney disease (CKD).  

Authors made a great effort to review the current literature on the topic.

Here are my comments,

1. Please disclose the search terms used to review the literature.

2. Authors should consider including the effects of the fiber and gut microbiota in the dialysis population.

3. Some literature on the topic, such as the study by Xu H et al., is omitted in the manuscript (PMID: 25280496), a study by Salmean at al., (PMID: 25446837)

Negative studies not mentioned in the article (PMID: 28582807)

Please have the manuscript reviewed by an English language expert before resubmission.

Author Response

Reviewer 3

Camerotto et al. have written a commentary on the relationship between dietary fiber and gut microbiota in the prevention and management of chronic kidney disease (CKD).  Authors made a great effort to review the current literature on the topic. Here are my comments,

Please disclose the search terms used to review the literature.

Answer: the following sentence has been added at the end of the introduction (Page 1, lines 42-46):

We searched MEDLINE through 1 July 2019 using the search terms '(dietary fiber OR fiber$ OR fibre$ OR polysaccharides OR polymers OR carbohydrate$ OR dietary carbohydrate OR fermentable OR fructans OR oligofructose$ OR inulin OR psyllium OR lactulose) AND (chronic kidney disease OR CKD OR chronic renal failure OR CRF OR renal insufficiency OR hemodialysis OR haemodialysis OR dialysis)'

Authors should consider including the effects of the fiber and gut microbiota in the dialysis population.

Answer: the suggested topic is presented in the manuscript as follows:

Page 9, lines 328-330 “Recent data show that serum concentrations of p-cresol  and indoxyl sulfate  are reduced by oral p-inulin intake in CKD patients on hemodialysis [7].”

Page 9, lines 340-346: “In a retrospective, propensity score-matched study of CKD patients who received more than 3 months of a KA supplemented low-protein diet during the year preceding the start of dialysis, a lower long-term risk of all-cause mortality (hazard ratio (HR) 0.77), major cardiac and cerebrovascular events (HR 0.86), and infection-related death (HR 0.76) was observed compared to patients on a different non-specified diet [Yen CL, 2018]. Bellizzi et al. [2015] confirmed that a very low-protein diet supplemented with amino acids and KA during CKD does not increase mortality in the subsequent renal replacement treatment period.”

In addition, a dedicated section was added (section 6, page 11, lines 426-446):

“Fiber in dialysis. The effects of fiber intake in dialysis patients has been assessed and reviewed by Sirich and collaborators [Sirich CJSN 2014; Sirich Semin Dial 2015]. In dialysis patients, an increase in dietary protein intake is suggested by current guidelines, between 1.1 g/kg/day [KDOQI – Kopple 2001] and 1,2 g/kg/day [EBPG – Fouque 2007]. This may lead to an increased production of protein-derived uremic toxins produced by the colon microbiota, which may be responsible for symptoms and increased cardiovascular morbidity. This effect may be offset by an increased fiber intake, as demonstrated by a study in hemodialysis patients, who showed significantly reduced plasma levels of indoxyl sulfate after an increase in dietary fiber in the form of high amylose corn starch for 6 weeks [Sirich 2014]. Another group [Meijers 2010] previously showed that increasing dietary fiber (oligofructose-enriched inulin for4 weeks) reduced the production of p-cresol in hemodialysis patients. The proposed protective mechanism of fiber is related to an increased delivery of SCFAs to the colon, providing energy and allowing microbes to incorporate dietary protein for growth rather than producing uremic toxins from protein breakdown [Nordgaard 1995].

Among non-diabetic peritoneal dialysis patients, fiber intake was inversely associated with all-cause mortality, with a 13% reduction of mortality for each 1g/day of increased fiber intake [Xu 2019]. Moreover, in another recent prospective cohort study of 219 prevalent hemodialysis patients, dietary fiber intake was associated with less inflammation, less myocardial hypertrophy, injury, and a reduced risk of major adverse cardiovascular events [Wang 2019].

Currently, there is no indication on the ideal fiber intake for hemodialysis patient, but the estimated intake is only 11-12 g/day, compared to the suggested intake for the general population of 25 g/day for women and 38 g/day for men [Sirich 2015].”

Some literature on the topic, such as the study by Xu H et al. 2014, is omitted in the manuscript (PMID: 25280496), a study by Salmean at al., (PMID: 25446837)

Negative studies not mentioned in the article (PMID: 28582807)

Answer: The following sentence was added (page 11, lines 423-425): “Considering the association of fiber and hard outcomes, high dietary total fiber intake has been linked with better kidney function and a lower risk of inflammation and mortality in CKD patients of North America [Krishnamurthy 2012] and Northern Europe [Xu 2014].“

In addition, the following sentence was added (page 11, lines 404-406): “Remarkably, fiber supplementation in CKD patients was shown to lower plasma p-cresol by 20% to 37% (in participants with elevated compliance) [Salmean 2015].”

Regarding the need to cite negative studies, the one suggested by the reviewer (PMID: 28582807) corresponds to the article by Lu et al. Dietary fiber intake is associated with chronic kidney disease (CKD) progression and cardiovascular risk, but not protein nutritional status, in adults with CKD. Asia Pac J Clin Nutr. 2017;26(4):598-605. However, different from what can be inferred from the title, the conclusions state that “Our results suggest that increasing fiber intake can retard the decrease in the eGFR; can reduce the levels of proinflammatory factors, indoxyl sulfate, and serum cholesterol; and is negatively associated with cardiovascular risk, but does not disrupt the nutritional status of patients with CKD.” Thus, this is not a negative study and we decided not to include it in the manuscript.

Please have the manuscript reviewed by an English language expert before resubmission.

Answer: the manuscript has been extensively reviewed, although not by an English-speaking person. We hope it now meets the standards of the Journal.

Reviewer 4 Report

I read with interest the article entitled “Dietary fiber and gut microbiota in renal diets” submitted for publication in Nutrients. This manuscript is relevant and appropriate to this journal as it summarizes the information regarding dietary fiber, definitions, types, and then possible outcomes in chronic kidney disease (CKD). I hope the authors will find the following comments helpful in revising their manuscript:

Overall comments

·         Throughout the manuscript, the authors fail to provide references.

·         The use of quotes is unnecessary in most instances.

Abstract

·         I would add a sentence or two on how we should focus on the viscosity and fermentability of dietary fibers instead of total dietary fiber and their current classification (i.e., soluble vs. insoluble).

Introduction

·         The authors only mention dietary protein, but dietary potassium and phosphorus restrictions are a big barrier for dietary fiber intake, which is the focus of the manuscript.

·         Line (L) 30: provide reference.

Body of the manuscript

·         L39-40: reference

·         Table 1: Are there any other guidelines, besides KDOQI that may have a recommendation for dietary fiber, such as the European Best Practice? I suggest also including the data by Beto et al. Medical Nutrition Therapy in Adults with Chronic Kidney Disease: Integrating Evidence and Consensus into Practice for Generalist Registered Dietitian Nutritionist. Journal of the Academy of Nutrition and Dietetics. Volume 114, Issue 7, July 2014, Pages 1077-1087.

·         L51-53: reference.

·         Consider including the definition for dietary fiber from the Codex Alimentarius and/or the US National Academy of Sciences and Engineering (former Institute of Medicine). Additionally, the US utilizes this definition.

·          L63: associated substances do not need quotes.

·         Table 2:

o   Can the authors provide evidence that cellulose is water-soluble? In most of the literature is the one type of fiber that is always listed as insoluble. Also, fermentability. It is my understanding that less than 5% is fermented and therefore in some studies it is used as a negative control.

o   For sources of inulin and fructans, are the cereal grains sources because it they are added or are they true sources of inulin/fructans?

o   I’m not sure if colled starches is a term, did the authors meant cooked and cooled?

·         L69: the period after (MU>3) is not needed.

·         L69-71: for reference, the adequate intake for the US was based on the cholesterol-lowering effect.

·         L77: the “and” between classification and that is not needed.

·         Table 3: it is stated that a better classification would be provided by including viscosity, fermentability, and bulking effect. However, bulking effects is not included in the table, water solubility is rather included (which was already mentioned in Table 2). Also, add a comma between cellulose and lignans as they are not the same.

·         L87-88: Reference.

·         L88-89: Reference.

·         L90-91: Reference.

·         L95-104: These sentences can be consolidated into a paragraph; the sentences by themselves are not justified.

·         L109: I suggest changing population to individuals.

·         Figure 1: It is confusing that the line between the “Fiber fermentation” box is only going to bulking effect. Was this intentional? Also, only fermentation is mentioned, what about viscosity?

·         L117-119: Reference.

·         L124: period after reference 19.

·         L124-125: Reference.

·         L127-129: Reference.

·         What is EPIC? (L132). Also for this study, did the study included participants with CKD?

·         I would suggest using kidney instead of renal. It is something that KDIGO has recommended to facilitate understanding in kidney disease terminology (i.e., end-stage kidney disease, kidney function, etc.) (L140, 148).

·         L152: capitalize K.

·         L152-153: Reference.

·         L153-155: add Huang et al reference.

·         L157: include 95% CI.

·         What is the connection between dietary fiber and RCC? Do the causes of this type of cancer are mostly lifestyle?

·         L168-170: Reference.

·         L176: change “reduce” to “reduced”

·         L 180: change “sugar” to “carbohydrates”

·         L 183-185: reference.

·         Was there no outcome at all from reference 26?

·         L192: please review paper by Sender et al. Revised estimates for the number of human and bacteria cells in the body.  PLOS Biology 2016 (here is a comment about this: https://www.nature.com/news/scientists-bust-myth-that-our-bodies-have-more-bacteria-than-human-cells-1.19136 )

·         L198-200: Reference.

·         L200-202: Reference.

·         L202-203: Reference.

·         L205;207: Reference.

·         L208-210: Reference.

·         L210-211: Reference.

·         L214-216: Reference.

·         L220: changes NANES to NHANES.

·         L222: modify renal to kidney.

·         L225-226: this sentence seem to be a conclusion.

·         L228: precursors of LPS? Can the authors expand on this? Or do they only mean bacterial translocation or products of bacterial lysis translocation?

·         L234: change trough to through

·         For study referenced in L240-241, was it 27g supplemented or complemented to a median intake of 27g/d?

·         L246: consider modifying “has been collected” to “was collected”

·         L249-251: include 95% CI.

·         L259: add that dietary protein restriction…

·         L261-262: Reference.

·         L262-264: Reference.

·         L267: SCFA abbreviation was used before.

·         L269-271: Reference.

·         L277: is it restored though? Few studies have really assessed changes in the composition of the microbiota. Usually a few bacterial general are mentioned, but overall modification is usually lacking.

·         L282: what does modest mean?

·         L309: is the low-protein diet in Italy animal-based? Or a few animal products are included with the majority being plant-based foods?

·         L318-321: Reference.

·         L344: is there evidence for a different bioavailability of potassium?

Conclusions

·         L356: use CKD.

·         I would recommend adding something related to the source of dietary fiber and characteristics (i.e., viscosity and fermentability)

·         Nothing is mentioned about the microbiota and dietary fiber.

Author Response

Reviewer 4

I read with interest the article entitled “Dietary fiber and gut microbiota in renal diets” submitted for publication in Nutrients. This manuscript is relevant and appropriate to this journal as it summarizes the information regarding dietary fiber, definitions, types, and then possible outcomes in chronic kidney disease (CKD). I hope the authors will find the following comments helpful in revising their manuscript:

Overall comments

Throughout the manuscript, the authors fail to provide references.

Answer: the number of references has been substantially increased

The use of quotes is unnecessary in most instances.

Answer: we could not understand which quotes the reviewer is referring to and therefore we were unable to correct them. We hope that despite their persistence the reviewer will be satisfied with the revision of the manuscript

Abstract

I would add a sentence or two on how we should focus on the viscosity and fermentability of dietary fibers instead of total dietary fiber and their current classification (i.e., soluble vs. insoluble).

Answer: the following sentence has been added in the abstract: “Fibers have been identified based on aqueous solubility, but a more complete classification is now used, also considering viscosity, fermentability, and bulking effect in the colon.”

Introduction

The authors only mention dietary protein, but dietary potassium and phosphorus restrictions are a big barrier for dietary fiber intake, which is the focus of the manuscript.

Answer:  we agree with the reviewer, but using a correct dietary counseling we can reduce the effective potassium and phosphorus load without limiting dietary fiber intake. We added the following sentence to the introduction (page 2, lines 36-38): “Although dietary phosphorus restrictions may be a barrier to an increased dietary fiber intake, using a correct dietary counseling we can reduce the effective potassium and phosphorus load without limiting fibers [Cupisti 2018, D’Alessandro 2015].”

Line (L) 30: provide reference.

Answer: four references were added: [Liabeuf 2010, Wu 2011, Wu 2012, Salmean 2015].

Body of the manuscript

L39-40: reference

Answer: done [Marlett 2002]

Table 1: Are there any other guidelines, besides KDOQI that may have a recommendation for dietary fiber, such as the European Best Practice? I suggest also including the data by Beto et al. Medical Nutrition Therapy in Adults with Chronic Kidney Disease: Integrating Evidence and Consensus into Practice for Generalist Registered Dietitian Nutritionist. Journal of the Academy of Nutrition and Dietetics. Volume 114, Issue 7, July 2014, Pages 1077-1087.

Answer: because there are no specific recommendations for CKD patients in nephrology guidelines, the following changes were done. Table 1: the legend was changed specifying that the suggested amount of fiber was derived from guidelines for the general population and that no specific recommendation is provided for CKD patients by the available nephrology guidelines. In the text, the following sentence and the Beto et al 2014 reference were added (page 2, lines 50-54): “The nutrition nephrology guidelines KDOQI (Kidney Disease Outcomes Quality Initiative) [8], KDIGO (Kidney Disease Improving Global Outcomes) [9], and EBPG (European Best Practice Guidelines) [Fouque 2007] do not provide specific guidance for CKD patients. Beto et al [2014] indicate for all CKD stages the same amount suggested for the general population.”

      L51-53: reference.

Answer: The Institute of Medicine reference was added (Trumbo et al. 2002)

Consider including the definition for dietary fiber from the Codex Alimentarius and/or the US National Academy of Sciences and Engineering (former Institute of Medicine). Additionally, the US utilizes this definition.

Answer: the following sentence was added (page 2, lines 77-85): “The US CODEX Alimentarius Commission, which sets international guidance standards for food, introduced the following definition of dietary fiber in 2009: carbohydrate (CHO) polymers with ten or more monomeric units, which are not hydrolyzed by the endogenous enzymes in the small intestine of humans and belong to the following categories: a) edible CHO polymers naturally occurring in the food as consumed; b) CHO polymers, obtained from food raw material by physical, enzymatic, or chemical means; and c) synthetic CHO polymers. [Lupton 2009]. It was also indicated that individual countries can decide whether they accept oligosaccharides with a degree of polymerization (DP) from 3 to 9 (included) as being fiber, raising some uncertainty regarding a correct and univocal definition.”

L63: associated substances do not need quotes.

Quotes were deleted

Table 2:

Can the authors provide evidence that cellulose is water-soluble? In most of the literature is the one type of fiber that is always listed as insoluble. Also, fermentability. It is my understanding that less than 5% is fermented and therefore in some studies it is used as a negative control.

Answer: this is a difficult question to address and we would not introduce a complex discussion in the manuscript. For a direct answer to the reviewer we refer to the following paper:

Meine N, Rinaldi R, Schüth F. Solvent-free catalytic depolymerization of cellulose to water-soluble oligosaccharides. ChemSusChem. 2012 Aug;5(8):1449-54. doi: 10.1002/cssc.201100770.

Cellulose shows very low reactivity in chemical or enzymatic hydrolyses due to its highly hydrogen‐bonded supramolecular structure, which limits access of reactants and catalysts to biopolymer chains and makes the substrate insoluble in conventional solvent. However, the human intestine is able to transform cellulose so that it becomes water soluble. Impregnation of cellulosic substrates with catalytic amounts of a strong acid (e.g., H2SO4, HCl) is a highly effective strategy for minimizing the contact problem commonly experienced in mechanically assisted, solid‐state reactions. Milling the acid‐impregnated cellulose fully converts the substrate into water‐soluble oligosaccharides within 2 h.

Accordingly, we changed the table, indicating the variable nature of cellulose.

For sources of inulin and fructans, are the cereal grains sources because it they are added or are they true sources of inulin/fructans?

We found information in the following paper:

Verspreet J et al. Cereal grain fructans: Structure, variability and potential health effects. Trends in Food Science & Technology 2015, 43, 32-42. Cereals are true sources of inulin/fructans  and they may have a profound impact on colon health. We added in the table that fructans are 2-3% w/w.

I’m not sure if colled starches is a term, did the authors meant cooked and cooled?

Answer: Yes, thank you for pointing this out. We corrected the table accordingly

L69: the period after (MU>3) is not needed.

Answer: we respectfully disagree. To make it more clear that we are dealing with two different sentences, we moved the second one to a new line.

L69-71: for reference, the adequate intake for the US was based on the cholesterol-lowering effect.

Answer: this concept was added to the sentence

L77: the “and” between classification and that is not needed.

Answer: The sentence was corrected, as suggested

Table 3: it is stated that a better classification would be provided by including viscosity, fermentability, and bulking effect. However, bulking effects is not included in the table, water solubility is rather included (which was already mentioned in Table 2).

Answer: to clarify, the sentence at page 4, lines 100-102 was changed as follows: “… so a different classification according to main characteristic of viscosity, fermentability and bulking effect in the colon, which is related to water solubility and swelling capacity, has been proposed”

Also, add a comma between cellulose and lignans as they are not the same.

Answer: done

L87-88: Reference. Answer: Slavin 2013 L88-89: Reference. Answer: Lattimer 2010 L90-91: Reference. Answer: McRorie 2015 L95-104: These sentences can be consolidated into a paragraph; the sentences by themselves are not justified. Answer: done. L109: I suggest changing population to individuals. Answer: done. Figure 1: It is confusing that the line between the “Fiber fermentation” box is only going to bulking effect. Was this intentional? Also, only fermentation is mentioned, what about viscosity?

Answer: No, it was not intentional. We believe that the arrow points to all the indicated consequences of fiber fermentation, not only to the bulking effect. Viscosity was not included in the figure, because it is mainly fermentation that will determine SCFA production, although viscosity may influence the fermentation ability of fiber.

L117-119: Reference. Answer: Thompson 2010 L124: period after reference 19. Answer: done. L124-125: Reference. Answer: Cano 2007, Cupisti 2013. L127-129: Reference. Answer: Slavin 2005. What is EPIC? (L132). Also for this study, did the study included participants with CKD?

Answer: European Prospective Investigation into Cancer and Nutrition. (added) 

I would suggest using kidney instead of renal. It is something that KDIGO has recommended to facilitate understanding in kidney disease terminology (i.e., end-stage kidney disease, kidney function, etc.) (L140, 148).

Answer: we respectfully point out that the term renal is widely used in Europe. We also expect that this article will be read by healthcare professionals, who certainly understand the word.

L152: capitalize K. Done L152-153: Reference. Answer: Wong 2017 L153-155: add Huang et al reference. Done L157: include 95% CI. Done What is the connection between dietary fiber and RCC? Do the causes of this type of cancer are mostly lifestyle?

Answer: Lifestyle is only one of the components increasing risk of kidney cancer. However, The dietary fiber intake, especially vegetable and legume fiber, may be associated with reduced RCC risk (Huang 2014).

L168-170: Reference. Answer: Slavin 2013 L176: change “reduce” to “reduced”. Done L 180: change “sugar” to “carbohydrates”. Done L 183-185: reference. Answer: Carvalho 2019 Was there no outcome at all from reference 26?

Answer: Yes, they were renal outcomes (albuminuria, proteinuria, changes in estimated glomerular filtration rate, dialysis) in patients with diabetes. The sentence was corrected, including the information.

L192: please review paper by Sender et al. Revised estimates for the number of human and bacteria cells in the body. PLOS Biology 2016 (here is a comment about this: https://www.nature.com/news/scientists-bust-myth-that-our-bodies-have-more-bacteria-than-human-cells-1.19136 )

Answer: Done. The sentence now reads as follows (page 7, lines 229-233): “Dietary fiber is the key substrate for microbial growth of bacteria living in the gut (prebiotic action). The gut microbiota is a true ecosystem made up by an estimated number of 3.8x1013 microorganisms living throughout the human body of a 70 kg individual [Sender 2016]. Thus, the number of bacteria in the body is of the same order as the number of human cells (3.0x1013), and their total mass is about 0.2 kg.”

L198-200: Reference. Answer: Slavin 2013 L200-202: Reference. Answer: Nicholson 2012 L202-203: Reference. Answer: Cummings 1981 L205;207: Reference. Answer: Wong 2006 L208-210: Reference. Answer: Huang 2017 L210-211: Reference. Answer: Koh 2016 L214-216: Reference. Answer: Krishnamurthy 2012 L220: changes NANES to NHANES. Done L225-226: this sentence seems to be a conclusion.

Answer: correct. We removed the sentence

L228: precursors of LPS? Can the authors expand on this? Or do they only mean bacterial translocation or products of bacterial lysis translocation?

Answer: we clarified the sentence (page 8, line 269-270): “… as well as products of bacterial lysis translocation, i.e. precursors for lipopolysaccharides (LPS), …”

L234: change trough to through. Done For study referenced in L240-241, was it 27g supplemented or complemented to a median intake of 27g/d?

The paper by Chiavaroli et al, a metanalysis, does not clearly specify how the fiber dose was determined in the individual studies. It appears that the dose includes dietary fiber and supplements. The sentence has been changed to clarify.

L246: consider modifying “has been collected” to “was collected”. Done L249-251: include 95% CI.

Answer: the following sentence was added (page 8, lines 391-302): “Odds ratios for participants in the highest compared with the lowest tertile of fiber from vegetables was 0·63 (95 % CI 0·43, 0·93) and from legumes it was 0·68 (95 % CI 0·47, 0·98)

L259: add that dietary protein restriction… Done L261-262: Reference. Answer: Ramezani 2014 L262-264: Reference. Answer: Aronov 2011 L267: SCFA abbreviation was used before. Corrected L269-271: Reference. Answer: Hobby 2019 L277: is it restored though? Few studies have really assessed changes in the composition of the microbiota. Usually a few bacterial general are mentioned, but overall modification is usually lacking.

Answer: The term “restoring”  has been replaced by “ modifying”  

L282: what does modest mean? Answer: mild L309: is the low-protein diet in Italy animal-based? Or a few animal products are included with the majority being plant-based foods?

Answer:  Animal-based low protein diet means that the proteins present in the diet are mostly from animal source : that is regular cereals and legumes are excluded and substituted by high-energy protein-free products (see old references 41,42,46,47) . The sentence was modified to clarify the concept (page 10, lines 373-3 77):  “ In Italy, an animal-based low protein diet includes the use of special low protein foods, formulated to give high number of calories with negligible content of nitrogen, potassium and phosphorous. In other words, the proteins in the diet are mostly from animal source, because cereals and legumes are excluded and substituted by high-energy protein-free products. These are starch-made substitutes of…”

L318-321: Reference.

Answer: we don’t have a literature reference for this sentence. The information is derived from the product information of protein free foods. Industries are:

Schär Corporate Communications & Nutrition Service, Dr. Schär AG/SPA, Winkelau 9, 39014 Burgstall/Postal (BZ) Italy Heinz Italia SpA Medical Food, Aproten, Altopascio, Lucca Italy

        L344: is there evidence for a different bioavailability of potassium?

Answser: thank you for pointing this out. We agree that potassium is fully absorbed in the intestinal proximal tract without any significant difference among the various potassium sources. Thus, potassium was removed from the sentence. 

Conclusions

L356: use CKD. Answer: done I would recommend adding something related to the source of dietary fiber and characteristics (i.e., viscosity and fermentability).

Answer: The following sentence was added to the conclusion: “Fibers are a diversified family of compounds, classified considering viscosity, fermentability, and bulking effect in the colon, which is related to aqueous solubility. The main sources of dietary fiber are the cereal grains, starchy endosperm, fruits and vegetable cell walls, green banana, psyllium, legumes, guar gum, beetroot, rice endosperm, chicory root, Jerusalem artichoke, onion, polymers derived by hydrolysis from polysaccharides, red algae, and fungi.”

Nothing is mentioned about the microbiota and dietary fiber.

Answer: the following sentence was added: “Dietary fiber can modulate the intestinal microbiota metabolism of CKD patients, reducing the bacterial generation of uremic toxins and potentially improving CKD progression and, even in dialysis patients, improving uremic symptoms and comorbidities.”

Round 2

Reviewer 2 Report

Table 2: As mentioned previously, I feel that the Tables show some inaccuracies. I would recommend to specify:

Cellulose, hemicellulose: The authors write in the column “Water solubility”: “- or ++”. Please specify, which fiber relates to “-“ and which to “++”. The authors also write in the column “Viscosity”: “Varies with source”. Again, please specify. Mannans, heteromannans: Again, the authors write in the column “Water solubility”: “++/+++”, in the column “Viscosity”: "-/+++” and in the column “Fermentability”: “+ /++”. Again, please specify, what applies to which specific fiber. Resistant Starch MU ≥ 10: Again, please specify in the column “Fermentability” which type relates to “+” and which type to “++”

Please also add references for the Tables 2 and 3 as well as for Figure 1.

I also mentioned in my first report that I do not think that it is correct to discuss if fibers can prevent CKD. They may lower the risk for the development, however, the authors also mention a disease prevention. However, even after the revision, the authors still use the same wording (e.g. lines 15, 38, 328).

Author Response

Reviewer 2

Comments and Suggestions for Authors

Table 2: As mentioned previously, I feel that the Tables show some inaccuracies. I would recommend to specify:

Cellulose, hemicellulose: The authors write in the column “Water solubility”: “- or ++”. Please specify, which fiber relates to “-“ and which to “++”. The authors also write in the column “Viscosity”: “Varies with source”. Again, please specify. Mannans, heteromannans: Again, the authors write in the column “Water solubility”: “++/+++”, in the column “Viscosity”: "-/+++” and in the column “Fermentability”: “+ /++”. Again, please specify, what applies to which specific fiber. Resistant Starch MU ≥ 10: Again, please specify in the column “Fermentability” which type relates to “+” and which type to “++”

Answer: the suggested corrections were introduced in Table 2. In order to avoid confusion with fiber characteristics, the compounds were divided into different lines of the table. We thank the reviewer for pointing out the need for further changes. We believe that now the table is improved and more clearly readable.

Please also add references for Tables 2 and 3 as well as for Figure 1.

Answer: references form where data included in tables were derived were added, as suggested. References supporting the concepts expressed in Figure 1 are multiple and they are reported in the relative sections in the text. We added the wording “(for references, see text)” in the figure legend. As a comment to the text, we indicated the specific references and the Editor can decide to add those rather than the wording “see text”  

I also mentioned in my first report that I do not think that it is correct to discuss if fibers can prevent CKD. They may lower the risk for the development, however, the authors also mention a disease prevention. However, even after the revision, the authors still use the same wording (e.g. lines 15, 38, 328).

Answer: we apologize for missing this request. The text has now been changed accordingly in the three indicated sections of the manuscript

Reviewer 4 Report

I thank the authors for taking my suggestions into consideration when revising their manuscript. I only have a couple of suggestions: 

Abstract 

Line (L) 19: use CKD.  l16-18: is that classification already used in Europe or it is suggested by the body of evidence linking it to a physiological benefit. At least in the US it is not used. 

Introduction

L34-36: I would suggest modifying the sentence saying that nutritional counseling can help individuals limit dietary phosphorus and potassium, while maintaining or increasing dietary fiber intake.  I am not sure why a methodology is included (L40-44) as this is not a systematic review and it is rather a narrative review. 

Body

Table 1 there is an extra bullet with no text.  Table 2: I disagree with the authors in regards to cellulose. The article they provided was an study adding H2SO4, HCl to depolymerize cellulose and even high temperatures were applied, which does not happen in vivo. A variety of studies utilize cellulose as a negative control in fiber supplementation studies due to the water insolubility and non-fermentability nature of the fiber. Therefore, I would strongly suggest to change the solubility of cellulose (not hemicelullose) to negative. Alternatively, the authors can separate cellulose and hemicellulose in their table to not confuse the audience.  L317-326: I would recommend adding the 95% CI to all the HR provided

Author Response

Reviewer 4

Comments and Suggestions for Authors

I thank the authors for taking my suggestions into consideration when revising their manuscript. I only have a couple of suggestions:

Abstract

Line (L) 19: use CKD. Answer: Done

L16-18: is that classification already used in Europe or it is suggested by the body of evidence linking it to a physiological benefit. At least in the US it is not used.

Answer: The most common definition of fibre dates back more or less to 2008 with the Codex Alimentarius, with a general global agreement. In 2010 the European Food Safety Authority (EFSA) remarked that ‘the terms “soluble” and “insoluble” have been used in the literature to classify dietary fibre in the attempt to link different physical-chemical properties of fibre components to different physiological effects. The classification by water solubility is method-dependent, does not always predict physiological effects and is not directly related either to fermentability or to the profile of SCFA produced with fermentation. Thus a new classification was proposed based on the main characteristics of dietary fibre that is viscosity in solution and/or in the digestive tract, fermentability in the colon (possibly taking into account the rate of fermentation and SCFA profile) and bulking effect in the colon to better associate physical-chemical characteristics to the physiological aspects. [Alison M. Stephen, Nutrition Research Reviews 2017]

We are not aware of the extent of use of this more articulated classification in the US, but we believe that it should be diffused and this manuscript could be a way to do so. Because it is difficult to specify these concepts in the abstract, in order to address the reviewer’s comment we decided to avoid the word “classification” and we changed the sentence as follows: “Fibers have been identified based on aqueous solubility, but other features, such as viscosity, fermentability, and bulking effect in the colon should be considered.”

Introduction

L34-36: I would suggest modifying the sentence saying that nutritional counseling can help individuals limit dietary phosphorus and potassium, while maintaining or increasing dietary fiber intake.  I am not sure why a methodology is included (L40-44) as this is not a systematic review and it is rather a narrative review.

Answer:  As suggested by the reviewer, we changed the sentence as follows: “ Nutritional counseling can help individuals limit dietary phosphorus and potassium load, while maintaining or increasing dietary fiber intake [8, 9]”. Regarding the methodology of the literature search, we agree with the reviewer and removed it. However, this was introduced following a specific request of another reviewer. Thus, we leave the final decision to the handling editor on whether to keep or delete this paragraph.

Body

Table 1 there is an extra bullet with no text.  Answer: corrected

Table 2: I disagree with the authors in regards to cellulose. The article they provided was a study adding H2SO4, HCl to depolymerize cellulose and even high temperatures were applied, which does not happen in vivo. A variety of studies utilize cellulose as a negative control in fiber supplementation studies due to the water insolubility and non-fermentability nature of the fiber. Therefore, I would strongly suggest to change the solubility of cellulose (not hemicelullose) to negative. Alternatively, the authors can separate cellulose and hemicellulose in their table to not confuse the audience.

Answer: Thank you for the request, which allows us to be more clear with the readers. We followed the reviewer’s advice and separated cellulose from hemicellulose in table 2 and 3, indicating that cellulose is not soluble

L317-326: I would recommend adding the 95% CI to all the HR provided.

Answer: The 95% CI to all the HR have been added, as follows: “In a retrospective, propensity score-matched study of CKD patients who received more than 3 months of a KA supplemented low-protein diet during the year preceding the start of dialysis, a lower long-term risk of all-cause mortality (hazard ratio (HR) 0.77, 95% confidence interval (CI) 0.70–0.84), major cardiac and cerebrovascular events (HR 0.86, 95% CI 0.78–0.94), and infection-related death (HR 0.76,  95% CI 0.67–0.87) was observed compared to patients on a different non-specified diet.”